

# S100A8/9 modulates perturbation and glycolysis of macrophages in allergic asthma mice

Xiaoyi Ji[1,2], Chunhua Nie[2], Yuan Yao[2], Yu Ma[1], Huafei Huang[2] and Chuangli Hao[1]

[1] Department of Respiratory Medicine, Children's Hospital of Soochow University, Suzhou, China
[2] Jiaxing Maternal and Child Health Hospital, Jiaxing, China

## ABSTRACT

**Background**. Allergic asthma is the most prevalent asthma phenotype and is associated with the disorders of immune cells and glycolysis. Macrophages are the most common type of immune cells in the lungs. Calprotectin (S100A8 and S100A9) are two pro-inflammatory molecules that target the Toll-like receptor 4 (TLR4) and are substantially increased in the serum of patients with severe asthma. This study aimed to determine the effects of S100A8/A9 on macrophage polarization and glycolysis associated with allergic asthma.

**Methods**. To better understand the roles of S100A8 and S100A9 in the pathogenesis of allergic asthma, we used ovalbumin (OVA)-induced MH-S cells, and OVA-sensitized and challenged mouse models (wild-type male BALB/c mice). Enzyme-linked immunosorbent assay, quantitative real-time polymerase chain reaction, flow cytometry, hematoxylin-eosin staining, and western blotting were performed. The glycolysis inhibitor 3-bromopyruvate (3-BP) was used to observe changes in glycolysis in mice.

**Results**. We found knockdown of S100A8 or S100A9 in OVA-induced MH-S cells inhibited inflammatory cytokines, macrophage polarization biomarker expression, and pyroptosis cell proportion, but increased anti-inflammatory cytokine interleukin (IL)-10 mRNA; also, glycolysis was inhibited, as evidenced by decreased lactate and key enzyme expression; especially, knockdown of S100A8 or S100A9 inhibited the activity of TLR4/myeloid differentiation primary response gene 88 (MyD88)/Nuclear factor kappa-B (NF-κB) signaling pathway. Intervention with lipopolysaccharides (LPS) abolished the beneficial effects of S100A8 and S100A9 knockdown. The observation of OVA-sensitized and challenged mice showed that S100A8 or S100A9 knockdown promoted respiratory function, improved lung injury, and inhibited inflammation; knockdown of S100A8 or S100A9 also suppressed macrophage polarization, glycolysis levels, and activation of the TLR4/MyD88/NF-κB signaling pathway in the lung. Conversely, S100A9 overexpression exacerbated lung injury and inflammation, promoting macrophage polarization and glycolysis, which were antagonized by the glycolysis inhibitor 3-BP.

**Conclusion**. S100A8 and S100A9 play critical roles in allergic asthma pathogenesis by promoting macrophage perturbation and glycolysis through the TLR4/MyD88/NF-κB signaling pathway. Inhibition of S100A8 and S100A9 may be a potential therapeutic strategy for allergic asthma.

Corresponding author
Chuangli Hao, hcl_md@sina.com

# INTRODUCTION

Asthma is a chronic inflammatory disease of the airways, clinically characterized by recurrent episodes of wheezing, chest tightness, or coughing, and its triggers include "extrinsic" (allergic asthma) and "intrinsic" (non-allergic asthma) (*Padem & Saltoun, 2019*; *Ioachimescu & Desai, 2019*). The prevalence of asthma ranges from 3.44% to 8.33% across all continents. Asthma sufferers have a reduced quality of life, and their families often account for a heavy healthcare cost burden (*Rabe et al., 2023*). Allergic asthma is the most common asthma phenotype, occurring on average at a younger age than endogenous asthma (*Schatz & Rosenwasser, 2014*) and is described as a chronic lung disease characterized by reversible airway obstruction, leading to airflow limitation and the manifestation of physiological symptoms (*Hough et al., 2020*). Macrophages, which are the most abundant immune cells in the lungs (approximately 70% of all immune cells), play an important role in allergic asthma caused by exogenous allergens (*Lee et al., 2015*; *Holt, 1986*). They are essential innate immune cells mainly classified as classical polarization (M1) or alternative polarization (M2). M1 macrophages are usually considered to have pro-inflammatory phenotypes with potent phagocytic and cytotoxic capacities, which are mainly defined by the expression of major compatibility complex II (MHCII), cluster of differentiation (CD) 14, CD80/CD86, CD38, and inducible nitric oxide synthase (iNOS) (*Saradna et al., 2018*). In contrast to M1 macrophages, M2 macrophages are considered to play a role in the abrogation of inflammatory and tissue repair pathways by expressing the cell surface biomarkers CD36, CD206, and CD163 (*Müller et al., 2007*; *Dewhurst et al., 2017*). However, a drastic change in the proportion of polarized macrophages was found in humans and mice with allergic asthma or lung inflammation; MHCII-hi macrophages (M1) and CD206$^+$ macrophages (M2) both increased, but interleukin (IL)-10$^+$ macrophages decreased (*Draijer et al., 2022*) suggesting a relationship between asthma and macrophages. CD86 and CD206 are key markers that distinguish between M1 and M2 macrophages, and their expression is markedly increased in allergic asthma (*Morsi et al., 2023*). Arginase 1 (Arg1), found in the inflammatory zone 1 (Fizz-1) is also a biomarker of M2 macrophages (*Xu et al., 2020*). In addition, activation of M2 macrophages promotes airway inflammation in asthma (*Zhong et al., 2023*). The homeostasis of macrophage perturbations is a key mechanism of airway inflammation in asthma; however, the exact mechanism remains poorly understood.

Over the past three years, immune metabolism studies in allergies have found that asthma is associated with increased aerobic glycolysis (*Goretzki et al., 2023*). A study of serum lactate acid levels in clinically stable patients with allergic asthma, healthy controls, and patients with chronic obstructive pulmonary disease (COPD) found that its content was substantially higher in patients than in healthy controls, which was also significantly higher than that in controls of patients with COPD (*Ostroukhova et al., 2012*) suggesting

an increase in glycolysis. Metabolism is critical for cellular activity as it provides energy and ATP, including the regulation of macrophages under homeostatic conditions and stress (*El Kasmi & Stenmark, 2015*). It has been found that under homeostatic conditions, the metabolic characteristics of macrophages are consistent with mitochondrial oxidative phosphorylation (OXPHOS) using glucose and oxygen (*Kelly & O'Neill, 2015*). *In vitro* studies have found that lipopolysaccharides (LPS), a Toll-like receptor 4 (TLR4) agonist, induced inflammatory macrophages decreased OXPHOS levels, along with an increase in the metabolism of glucose to lactate acid (*Palsson-McDermott et al., 2015*). In particular, there is evidence indicating that alveolar macrophages show a diminished response to IL-4 when recovering from *in vitro* incubation for 48 h. Additionally, the competitive glucose inhibitor 2-Deoxy-d-glucose (2-DG) significantly hinders the IL-4 induced up-regulation of Retnla, Arg1, and Chil2 (M2 macrophage biomarkers) in alveolar macrophages during *in vitro* incubation. This suggests that the lung environment plays a role in regulating metabolism, thereby influencing the polarization of alveolar macrophages (*Svedberg et al., 2019*).

S100A8 and S100A9 are two pro-inflammatory molecules belonging to the S100 family of calcium-binding proteins that normally form a heterodimeric complex (S100A8/A9, also known as calreticulin A and B) after being released from myeloid cells (*Vogl et al., 2006*). Their levels were significantly increased in the serum of patients with severe asthma compared to healthy controls (*Decaesteker et al., 2022*). *Lee et al. (2020)* found that in allergic asthmatic mice with type 2 airway inflammation, serum S100A8/A9 levels were correlated with lung function and airway hyperresponsiveness, implying that S100A8/A9 serves as a biomarker for asthma. Another study found that macrophages in the peripheral blood of patients with severe asthma expressed more S100A9 than those of patients with non-severe asthma (*Quoc et al., 2021*). A recent study reported that hexokinase (HK) 1 and glyceraldehyde-3-phosphate dehydrogenase (GAPDH) activities in macrophages from low-grade inflammation in humans and mice were reduced. HK1 interacting with S100A8/A9 can blockade glycolysis below GAPDH by nitrosylating GAPDH *via* nitric oxide synthase 2 (iNOS) (*De Jesus et al., 2022*). HK1 is an HK isoform and is one of the key enzymes involved in glucose phosphorylation (the first step in the glucose metabolism pathway), which includes HK1, HK2, and HK3 (*Yang et al., 2022*). It has been suggested that S100A8/A9 can regulate macrophage glycolysis in patients with allergic asthma. A study reported that S100A8/A9 complex played a key role in macrophage polarization to trigger inflammation in sepsis, as the endogenous ligand for TLR4, induced intracellular translocation of myeloid differentiation primary response gene 88 (MyD88) as well as nuclear factor kappa-B (NF-κB) activation to promote tumor necrosis factor (TNF)-α expression (*Vogl et al., 2007*). *Li et al. (2021b)* found that regulation of the TLR4/MyD88/NF-κB signaling pathway inhibited macrophage inflammation. LPS activates TLR4 and triggers MyD88 and TIR domain-containing adaptor, inducing interferon-β (TRIF) signaling cascades to induce inflammation (*Ciesielska, Matyjek & Kwiatkowska, 2021*). The metabolic enzyme ATP citrate lyase (ACLY) is a producer of citrate-derived acetyl-coenzyme A (CoA), which plays a critical role in supporting the pro-inflammatory response (*Santarsiero et al., 2021*). TLR4 activates ACLY to induce the inflammatory response (*Lauterbach et al., 2019*).

S100A8 and S100A9, the pro-inflammatory proteins targeting the TLR4, deserve further exploration for their roles in macrophage polarization and glycolytic metabolism.

Moreover, macrophage polarization is related to cellular metabolism and pyroptosis, and inhibition of glycolysis suppresses pyroptosis (*Zasłona et al., 2020*; *Aki et al., 2022*). S100A8 and S100A9 may be involved in the regulation of macrophage polarization, glycolytic metabolism, and pyroptosis in allergic asthma. Therefore, we established ovalbumin (OVA)-induced alveolar macrophage models to observe the effects of S100A8 and S100A9 on macrophage polarization. In addition, we established OVA-sensitized and -challenged mouse models to verify the protective effects of sh-S100A8 and sh-S100A9. This study aims to provide a scientific basis for exploring the effects of macrophage polarization and glycolytic metabolism on allergic asthma and provides new ideas for improving allergic asthma.

## MATERIALS & METHODS

### Cell culture and modeling

Mouse alveolar macrophages MH-S (iCell-m078, iCell Bioscience, Shanghai, China) were cultured in complete medium containing DMEM, 10% fetal bovine serum (13011-8611, Every Green, Taiwan), 0.05 mM β-mercaptoethanol (M6250, Sigma-Aldrich, St. Louis, MO, USA) and 1% penicillin/streptomycin (C0222, Beyotime, Suzhou, China). In addition, Lipofectamine 2000 (11668019, Invitrogen, Carlsbad, CA, USA) with sh-S100A8 or sh-S100A9 was incubated with cells ($2 \times 10^5$ cells/well) in 6-well plates for 48 h. Furthermore, macrophage polarization was induced for 8 h with OVA (*Wang et al., 2021*). Lipopolysaccharide (LPS) (100 ng/mL, HY-D1056, MedChemExpress, Boston, MA, USA) as a TLR4 agonist was incubated with MH-S cells for 8 h (*Alhouayek et al., 2013*) with PBS as a control. The cells were divided into Control, OVA, OVA + sh-S100A8, OVA + sh-S100A9, OVA + sh-S100A8 + LPS, and OVA + sh-S100A9 + LPS groups. Cell experiments of cells were repeated independently at least thrice.

### Animals with allergic asthma and grouping

Wild-type male BALB/c mice (approximately 19 g), aged 6–8 weeks, from Lingchang Biotech Co., Ltd, (Shanghai, China). The mice were kept in ventilated cages in a pathogen-free animal facility and provided free access to food and water. All animal protocols were performed by professionals blinded to the group assignment in compliance with the Guidelines for the Humane Treatment of Laboratory Animals and were approved by the Animal Experimentation Ethics Committee of Zhejiang Eyong Pharmaceutical Research and Development Center (approval number: ZJEY-20221205-02).

We injected mice with physiological saline, lentivirus-sh-S100A8/A9, or lentivirus-S100A9 ($1 \times 10^9$ viral particles per mouse) through the tail vein a week before OVA sensitization. The mice were sensitized and challenged with OVA (A5503, Sigma, Burlington, MA, USA) as previously described (*Li et al., 2021a*). Briefly, we injected mice intraperitoneally 10 μg of OVA in 1 mg aluminum hydroxide (239186, Sigma, Burlington, MA, USA) with 20 g Freund's adjuvant (77140, Thermo Fisher Scientific, Waltham, MA, USA) on days 0, 7 and 14. We subsequently challenged the mice with an OVA aerosol

using an ultrasonic nebulizer (Pari Proneb nebulizer, Midlothian, WA, USA) with a 1% (wt/vol) OVA solution in saline for 20 min on days 15–21, once a day. Additionally, we intraperitoneally injected 3-bromopyruvate (3-BP, 5 mg/kg/day) into the mice once every 2 days (*Fang et al., 2019*) to observe glycolysis.

Randomized allocation of animals using the random number table method. We divided 24 mice into four groups to observe the effects of S100A8 and S100A9 knockdown on allergic asthma (six mice per group): negative control (NC), OVA, OVA + sh-S100A8, and OVA + sh-S100A9 groups. Additionally, 24 mice were divided into four groups to observe glycolysis (six mice per group): OVA, OVA + 3-BP, OVA + OE-S100A9, and OVA + S100A9 + 3-BP. In animal experiments, if the animals suffered infections and showed a sudden weight loss of more than 20% in one week, they were euthanized by $CO_2$ asphyxiation. In this study, the body weight, nutrition, and activity levels of the animals were normal. Therefore, none of these animals were excluded from the study. On day 22, a small-animal ventilator (flexiVent, SCIREQ, Montreal, Canada) was used to assess respiratory function in mice administered pentobarbital sodium for anesthesia. After mice were anesthetized with isoflurane on day 23, orbital blood was sampled to detect OVA-specific IgE and lactic acid. Following this, the mice were euthanized with carbon dioxide. Subsequently, bronchoalveolar lavage fluid (BALF) was collected for inflammatory cell counts, flow cytometry, and detection of inflammatory factors. After lavage, the lungs were isolated to fix for immunohistochemistry and histological observation or frozen at $-80\ °C$ for detection of gene and protein levels.

## Detection of mRNAs by quantitative real-time PCR

Through lysis and centrifugation, total RNAs of fresh cells or tissues (frozen at $-80\ °C$) were extracted using a total RNA small extraction kit (B618583-0100, Sangon, China). Total RNA was treated with RNase-free DNaseI. RNase-free water (20 μL; Am9932, Thermo Fisher Scientific, Waltham, MA, USA) was used to dissolve the total RNA, and an ultra-micro spectrophotometer (Nanodrop One, Thermo Fisher Scientific, Waltham, MA, USA) was used to determine its concentration and purity. Samples with ratios of 260/280 nm values between 1.9 and 2.1 and 260/230 nm values greater than 2.0, were used for subsequent experiments. HiFiScript cDNA Synthesis Kit (CW2569, CWBIO, Jiangsu, China) and SYBR Green qPCR Kit (11201ES08, Yeasen, Shanghai, China) were used for reverse transcription and stored at $-20\ °C$ pending analysis. Subsequently, quantitative real-time PCR (qRT-PCR) was performed using a LightCycler96 qRT-PCR instrument (Roche, Basel, Switzerland). The cycling conditions were determined according to the manufacturer's instructions. β-actin was used as an endogenous control. All data were processed by relative quantitative method ($2^{-\triangle\triangle Ct}$). The primer sequences are listed in Table 1. Genomic DNA contamination was not detected in the qRT-PCR products.

## Detection of inflammatory factors, lactic acid content, and glycolysis

Cell supernatant was used to detect the concentration of IL-1β (MM-0040M2, MEIMIAN, Jiangsu, China), IL-6 (MM-0163M2, MEIMIAN, Jiangsu, China), and TNF-α (MM-0132M2, MEIMIAN, Jiangsu, China) using ELISA. In addition, after testing respiratory

**Table 1  Primer sequences information.**

| Gene | 5′-Forward Primer-3′ | 5′-Reverse Primer-3′ | Sequence accession number | Amplicon length (bp) |
|---|---|---|---|---|
| Mouse S100A8 | AAATCACCATGCCCTCTACAAG | CCCACTTTTATCACCATCGCAA | NM_013650.2 | 165 |
| Mouse S100A9 | ATACTCTAGGAAGGAAGGACACC | TCCATGATGTCATTTATGAGGGC | NM_001281852.1 | 129 |
| Mouse iNOS | GGAGTGACGGCAAACATGACT | TCGATGCACAACTGGGTGAAC | NM_001313922.1 | 127 |
| Mouse IL-6 | TCTATACCACTTCACAAGTCGGA | GAATTGCCATTGCACAACTCTTT | NM_001314054.1 | 88 |
| Mouse Arg1 | TGTCCCTAATGACAGCTCCTT | GCATCCACCCAAATGACACAT | NM_007482.3 | 204 |
| Mouse IL-10 | CTTACTGACTGGCATGAGGATCA | GCAGCTCTAGGAGCATGTGG | NM_010548.2 | 101 |
| Mouse PDH | TGTGACCTTCATCGGCTAGAA | TGATCCGCCTTTAGCTCCATC | NM_008810.3 | 119 |
| Mouse HK2 | TGATCGCCTGCTTATTCACGG | AACCGCCTAGAAATCTCCAGA | NM_013820.4 | 112 |
| Mouse LDHA | CAAAGACTACTGTGTAACTGCGA | TGGACTGTACTTGACAATGTTGG | NM_001136069.2 | 148 |
| Mouse GAPDH | CGAGACACGATGGTGAAGGT | TGCCGTGGGTGGAATCATAC | NM_001411843.1 | 282 |
| Mouse IL-1β | GAAATGCCACCTTTTGACAGTG | TGGATGCTCTCATCAGGACAG | NM_008361.4 | 116 |
| Mouse Fizz1 | CCAATCCAGCTAACTATCCCTCC | ACCCAGTAGCAGTCATCCCA | NM_020509.4 | 108 |
| Mouse β-actin | GGCTGTATTCCCCTCCATCG | CCAGTTGGTAACAATGCCATGT | NM_007393.5 | 154 |

function, blood was collected *via* orbital blood collection, isoflurane-anesthetized mice were euthanized with $CO_2$, the lungs were lavaged twice with one mL of cold PBS, and BALF was collected. Serum OVA-specific IgE and lactic acid levels were detected using ELISA (MM-45386M2, MEIMIAN, Jiangsu, China) and a lactic acid content detection kit (BC2230, Solarbio, Beijing, China). Supernatant of BALF was counted to detect the concentration of IL-4 (MM-01065M2, MEIMIAN), IL-13 (MM-0173M2, MEIMIAN), TGF-β1 (MM-0135M2, MEIMIAN), TNF-α, and INF-γ (MM-0182M2, MEIMIAN). The extracellular acidification rate (EACR) kit (BB48311, BestBio, Shanghai, China), phosphofructokinase (PFK) kit (BC0530, Solarbio, Beijing, China), and hexokinase (HK) kit (BC0745, Solarbio, Beijing, China) were used to observe glycolysis in cells. Assays were performed using a multifunctional microplate reader (CMaxPlus, San Francisco, CA, USA).

## Inflammatory cell count

BALF cells were centrifuged (Cytospin 500, Sandton, UK) at 3,000 rpm for 5 min at 22 °C using cell centrifugation. Then, the cells on slides were stained with the Wright-Giemsa Staining kit (D010-1-2, Nanjing Jiancheng, China), air-dried, and fixed with Permount Mounting Medium (MM1411, MKBio, China). The slides were observed under a microscope, and at least 300 cells were counted for each preparation.

## Polarization and pyroptosis detection by flow cytometry

To confirm MH-S cells and lung macrophage polarization and pyroptosis were analyzed by flow cytometry. After processing the cells according to the grouping, MH-S cells were prepared into $2 \times 10^7$ cells/mL PBS suspensions. Subsequently, the FITC anti-F4/80 (ab60343, Abcam, Cambridge, UK), APC anti-CD86 (ab218757, Abcam, Cambridge, UK), and PE/Cy7 anti-mannose receptor [15-2] (CD206) (ab270682, Abcam, Cambridge, UK)

antibodies were incubated with MH-S cells at 4 °C for 15 min. Additionally, the BALF was centrifuged at 1,000 rpm for 5 min to collect the cells. Cells from BALF resuspended by PBS to $2 \times 10^7$ cells/mL and were incubated at 4 °C for 15 min with antibodies including Alexa Fluor® 700 anti- CD45 (157210, Biolegend, San Diego, CA, USA), FITC anti-F4/80, APC anti-CD86, PE/Cy7 anti-mannose receptor [15-2] (CD206), and PE anti-CD11b (ab25175, Abcam, Cambridge, UK). The gating strategy of BALF is described in Fig. S1. For pyroptosis, the FLICA 660 Caspase-1 (9122, ImmunoChemistry, Bloomington, MN, USA) and PI staining (556547, BD, Franklin Lakes, NJ, USA) were used to incubate the cells at 37 °C for 15 min. The percentage of cells that were double-positive for caspase-1 and PI was used to indicate pyroptosis. All cells were screened through a 200-mesh sieve and flow cytometry (NovoCyte, Agilent, Santa Clara, CA, USA) was used to detect the proportion of cells.

## Hematoxylin and eosin staining and immunohistochemistry

After euthanasia, the lungs were isolated from the mice, fixed with 4% paraformaldehyde for 24 h, and embedded in paraffin blocks. Subsequently, lung histopathology was performed using hematoxylin and eosin (HE) staining. Lung paraffin blocks were sliced into 4 μm sections, dewaxed, and hydrated. Finally, the sections were stained using an HE kit (C0105S, Beyotime, Shanghai, China) to observe morphology and cell morphology under a Nikon Eclipse Ci-L microscope (Tokyo, Japan).

## Western blot

Cell or tissue proteins were extracted using RIPA solution (P0013C, Beyotime, Shanghai, China) containing protease inhibitors (CW2200S, CWBIO, Beijing, China). After extraction and denaturation of total proteins, 10% gel electrophoresis was used to separate the proteins, and activated PVDF membranes were used for transfer. Subsequently, the blocked-membranes with 5% skim milk powder were incubated with primary antibodies such like anti-TLR4 (1:1000, 14358S, CST, Boston, MA, USA), anti-MyD88 (1:1000, ab219413, Abcam, Cambridge, UK), anti- TIR domain containing adaptor molecule 1 (TRIF) (1:3000, ab13810, Abcam, Cambridge, UK), anti-NF-κB (1:3000, 8242T, CST, Danvers, MA, USA), anti-IκBα (1:3000, 4814T, CST, Danvers, MA, USA), Anti-ATP citrate lyase antibody (ACLY) (1:10000, ab40793, Abcam, Cambridge, UK), phospho-ACLY (1:1000, 4331T, CST, Danvers, MA, USA), and anti- β-actin (1:20000, 81115-1-RR, Proteintech, Chicago, IL, USA) followed by corresponding secondary antibodies (1:6000, 7074/7076, CST, Danvers, MA, USA) at 25 °C for 2 h. The membrane was incubated with ECL reagents (610020-9Q, Qing Xiang, Shanghai, China) and visualized by ImageJ software.

## Statistical analysis

The statistical software SPSS 19.0 (IBM, Armonk, NY, USA) was used to analyze the data, and the continuous variables were presented as mean ± standard deviation. Data from multiple *in vivo* experiments were analyzed using one-way analysis of variance (ANOVA) with a *post-hoc* Tukey test. In cases where measurement data were not normally distributed,

the Kruskal–Wallis H test was used. Any $p$-value that resulted in lower than 0.05 was deemed statistically significant.

## RESULTS

### S100A8/9 knockdown inhibited polarization of ovalbumin-induced MH-S model

We examined the effects of S100A8 and S100A9 knockdowns on MH-S cell polarization and inflammation. As shown in Fig. S2, the S100A9 or S100A8 knockdown cell models were successfully established. The concentrations of the inflammatory factors, IL-1β, IL-6, and TNF-α were significantly increased in the OVA group ($p < 0.01$), while the knockdown of S100A8 or S100A9 decreased their concentrations ($p < 0.01$) (Figs. 1A–1C). We detected biomarkers mRNA of M1 macrophage (IL-6, IL-1β, and iNOS) and M2 macrophage (IL-10, Arg1, and Fizz1) and found that they were increased in the OVA group, except for IL-10 ($p < 0.01$), while knockdown of S100A8 or S100A9 decreased IL-6, IL-1β, iNOS, Arg1 and Fizz1 mRNA and increased IL-10 mRNA ($p < 0.01$) (Figs. 1D–1I). Furthermore, we detected the proportion of M1 (CD86+) and M2 (CD206+) cells using flow cytometry (Figs. 1J, 1K). In the OVA group, the proportion of M1 (CD86+) and M2 (CD206+) cells was increased, whereas S100A8 or S100A9 knockdown inhibited this increase ($p < 0.01$) (Figs. 1J, 1K). Additionally, LPS intervention antagonized the effects of S100A8 or S100A9 knockdown on OVA-induced MH-S cell injury models ($p < 0.05$) (Figs. 1A–1K). These results suggest that the knockdown of S100A8 or S100A9 has an inhibitory effect on OVA-induced macrophage polarization, whereas LPS is a TLR4 agonist that can partially antagonize this inhibitory effect.

### S100A8/9 knockdown inhibited pyroptosis and glycolysis of ovalbumin-induced MH-S model

We measured pyroptosis and found that the knockdown of S100A8 or S100A9 significantly reduced MH-S cell pyroptosis ($p < 0.05$), while LPS antagonized it ($p < 0.01$) (Figs. 1N, 1O). We also measured the EACR to observe glycolysis in MH-S cells. OVA intervention significantly decreased EACR at 1–63 min ($p < 0.01$) (Table 2). S100A8 knockdown significantly decreased ECAR of MH-S cells with OVA intervention, except at 1 and 36 min ($p < 0.05$) (Table 2). S100A9 knockdown significantly decreased ECAR of MH-S cells with OVA intervention at 9–63 min ($p < 0.05$) (Table 2). LPS intervention significantly counteracted the inhibitory effect of S100A8 knockdown ($p < 0.05$) and in OVA-inverted MH-S cells with S100A9 knockdown, LPS intervention significantly increased the ECAR at 9, 18, 27, 45, 54, and 63 min ($p < 0.05$) (Table 2). Furthermore, the concentrations of lactic acid, FPK, and HK were increased in MH-S cells treated with OVA intervention ($p < 0.01$) (Figs. 2A–2C). We also measured the mRNA levels of glycolysis-related genes. In OVA-induced MH-S cells, GAPDH, HK2, and LDHA mRNA levels significantly increased, whereas PDH mRNA levels decreased ($p < 0.01$) (Figs. 2D–2F). S100A8 or S100A9 knockdown inhibited GAPDH, HK2, and LDHA mRNA expression in OVA-induced MH-S cell injury models and increased PDH mRNA levels ($p < 0.01$); however, LPS intervention antagonized these effects ($p < 0.05$) (Figs. 2D–2G). The activation of

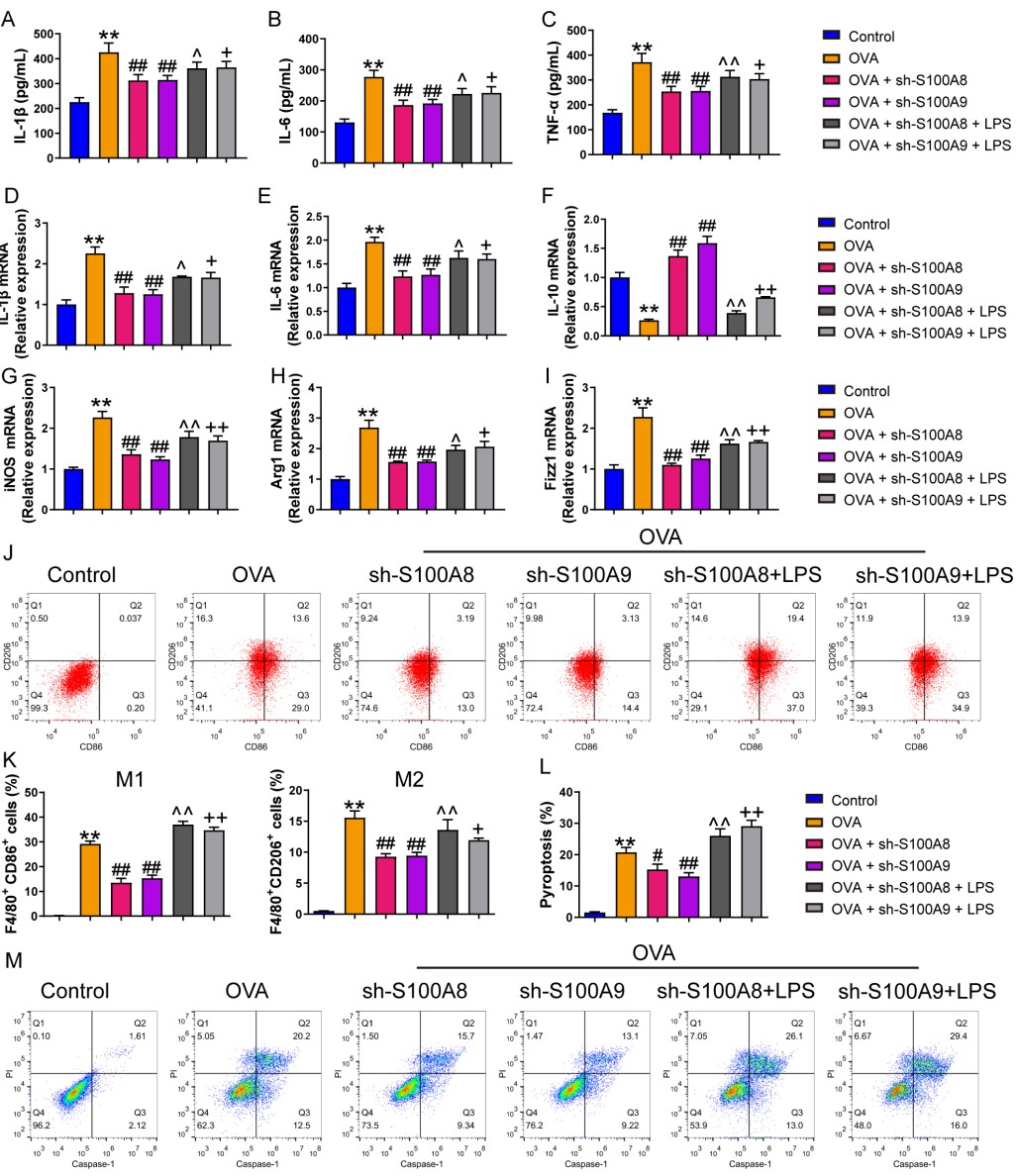

**Figure 1** **S100A8/9 knockdown inhibited polarization and pyroptosis of mouse alveolar macrophages with ovalbumin intervention.** Mouse alveolar macrophages (MH-S cells) treated with S100A8, S100A9, or lipopolysaccharide (LPS) were incubated with 40 M ovalbumin (OVA) for 8 h to establish alveolar macrophage inflammation models. MH-S cells were divided into six groups ($n = 3$ per group): control, OVA + sh-NC, OVA + S100A8, OVA + S100A8, OVA + S100A9, OVA + S100A8 + LPS, and OVA + S100A9 + LPS. In the cell supernatant, the concentrations of IL-1β (A), IL-6 (B), and TNF-α (C) were measured by ELISA. In MH-S cells treated with OVA, IL-1β (D) and IL-6 (E) mRNA levels were increased, but IL-10 (F) mRNA was decreased. In OVA-induced MH-S cells, the knockdown of S100A8 or S100A9 decreased the concentration and mRNA levels of IL-1β and IL-6, increased IL-10, and inhibited the mRNA expression of iNOS (M1 macrophage biomarker) (G), Arg1 (H), and Fizz1 (M2 macrophage biomarker) (I). All mRNAs were detected using quantitative real-time PCR. Mouse alveolar macrophage polarization (J and K) and pyroptosis (L and M) were analyzed by flow cytometry. In the OVA group, the proportion of M1 (CD86+) and M2 (CD206+) cells increased, whereas S100A8 and S100A9 knockdown inhibited this increase. Additionally, LPS intervention antagonized the effects of S100A8 and S100A9 knockdown in the OVA-induced MH-S cell injury models. 

**Figure 1 (…continued)**
(mean ± standard deviation) $^{**}p < 0.01$ compared to control group; $^{\#}p < 0.05$, $^{\#\#}p < 0.01$ compared to OVA group; $^{\wedge}p < 0.05$, $^{\wedge\wedge}p < 0.01$ compared to OVA + sh-S100A8 group; $^{+}p < 0.05$, $^{++}p < 0.01$ compared to OVA + sh-S100A8 group.

**Table 2  Effects of S100A8 and S100A9 knockdown on EACR in MH-S cells with OVA.** Extracellular acidification rate (EACR) was used to observe glycolysis in MH-S cells.

| Group | ECAR (mpH/min) | | | | | | | |
|---|---|---|---|---|---|---|---|---|
| | 1 min | 9 min | 18 min | 27 min | 36 min | 45 min | 54 min | 63 min |
| Control | 65.68 ± 4.11 | 44.62 ± 2.08 | 36.41 ± 3.2 | 54.32 ± 4.03 | 75.77 ± 5.58 | 50.75 ± 4.6 | 53.05 ± 2.95 | 41.28 ± 4.11 |
| OVA | 80.33 ± 7.74[**] | 65.67 ± 5.58[**] | 51.54 ± 4.62[**] | 76.06 ± 4.31[**] | 91.35 ± 4.69[**] | 70.73 ± 6.52[**] | 74.28 ± 7.37[**] | 50.17 ± 5.46[*] |
| OVA + sh-S100A8 | 75.53 ± 5.39 | 55.12 ± 4.29[#] | 40.99 ± 4.51[##] | 64.18 ± 3.7[##] | 85.03 ± 6.06 | 61.09 ± 4.89[#] | 64.38 ± 5.22[#] | 35.16 ± 3.33[##] |
| OVA + sh-S100A9 | 70.09 ± 5.87 | 35.35 ± 3.9[##] | 43.15 ± 3.66[#] | 59.86 ± 6.2[##] | 80.29 ± 6.72[#] | 55.41 ± 4.68[##] | 57.01 ± 3.96[##] | 42.37 ± 3.69[#] |
| OVA + sh-S100A8 + LPS | 86.75 ± 6.45[a] | 65.15 ± 6.07[a] | 55.28 ± 4.12[b] | 75.52 ± 4.57[b] | 96.47 ± 6.2[a] | 72.01 ± 3.76[b] | 75.4 ± 5.72[a] | 55.35 ± 4.62[b] |
| OVA + sh-S100A9 + LPS | 76.98 ± 7.2 | 64.25 ± 8.79[++] | 52.76 ± 5.52[++] | 72.71 ± 5.29[++] | 87.55 ± 5.98 | 73.78 ± 4.3[++] | 74.25 ± 5.57[++] | 50.84 ± 4.29[+] |

**Notes.**
[**] $p < 0.01$, compared to Control group.
[#] $p < 0.05$, compared to OVA group.
[##] $p < 0.01$, compared to OVA group.
[a] $p < 0.05$, compared to OVA + sh-S100A8 group.
[b] $p < 0.01$, compared to OVA + sh-S100A8 group.
[+] $p < 0.05$, compared to OVA + sh-S100A8 group.
[++] $p < 0.01$, compared to OVA + sh-S100A8 group.

the TLR4/MyD88/TRIF/NF-κB signaling pathway promotes glycolysis (*Li et al., 2021b*). Therefore, we detected the TLR4/MyD88/TRIF/NF-κB signaling pathway by Western blot. The levels of TLR4, MyD88, TRIF, NF-κB, and p-ACLY/ACLY in MH-S cells were increased and S100A8 or S100A9 knockdown decreased them ($p < 0.01$), while LPS intervention partially antagonized the inhibition of S100A8 or S100A9 knockdown on TLR4/MyD88/TRIF/NF-κB signaling pathway ($p < 0.05$) (Figs. 2H–2K, 2M, 2N). Furthermore, IκBα expression levels had the opposite trend to these proteins ($p < 0.05$) (Figs. 2L, 2N). We found that S100A8 or S100A9 knockdown inhibited macrophage pyroptosis and glycolysis through the TLR4/MyD88/TRIF/NF-κB signaling pathway.

## S100A8/9 knockdown improved respiratory function, lung tissue injury, and inflammation of BALF in ovalbumin-sensitized and challenged mice

We established a mouse model of allergic asthma using OVA to observe the protective effects of S100A8 and S100A 9 knockdown in the lungs post-OVA-challenged. As shown in Fig. S3, we successfully established OVA-sensitized and -challenged mice with S100A8 or S100A9 knockdown. To verify the role of S100A8 and S100A9 in allergic asthma mice, we assessed respiratory function, observed pathological damage to lung tissue, and measured the levels of inflammatory cells and cytokines in BALF. The respiratory function indicators tidal volume (TV), vital capacity (VC), expiratory volume (EV), minute ventilation volume (MV), forced expiratory volume in 0.1 s (FEV0.1), end inspiratory pause (EIP), peak expiratory flow (PEF), mid expiratory flow (EF50), and dynamic lung compliance (Cdyn) in OVA mice were markedly decreased while S100A8 and S100A9 knockdown reversed

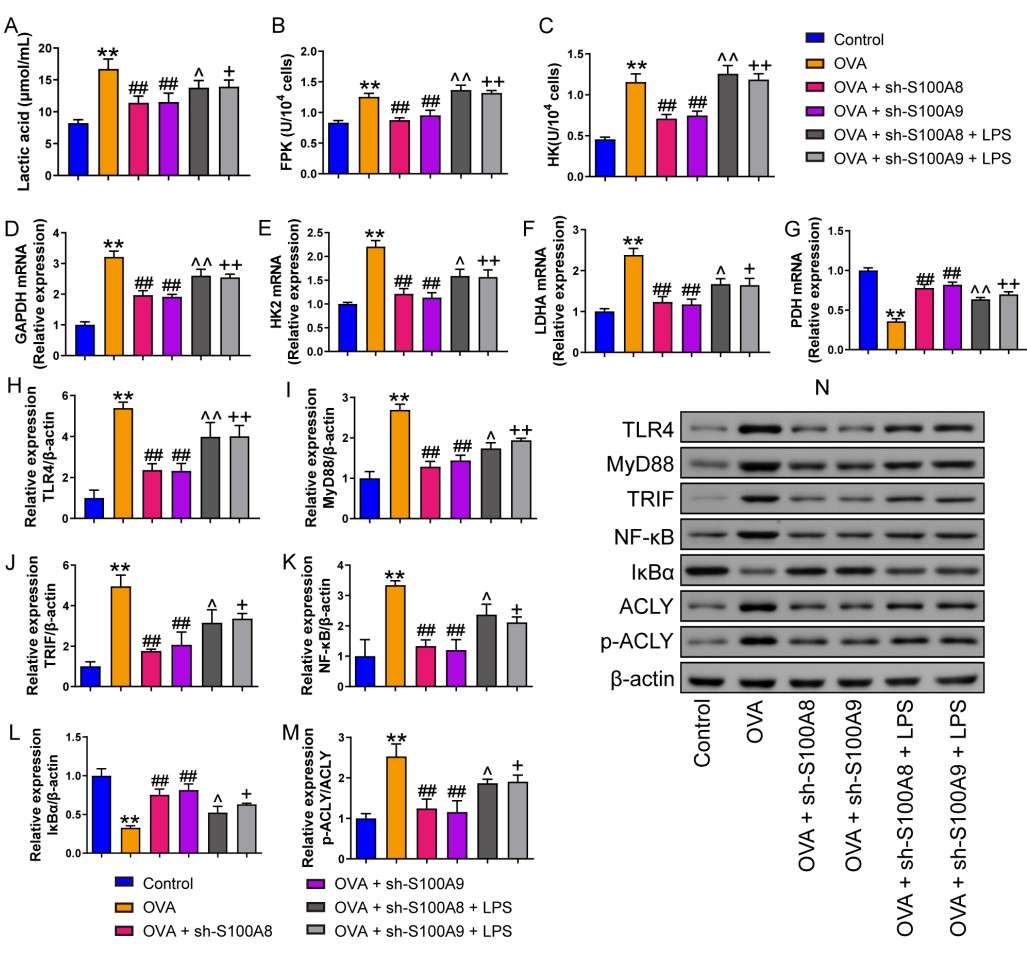

**Figure 2 S100A8/9 knockdown inhibited glycolysis of mouse alveolar macrophages with ovalbumin intervention.** Mouse alveolar macrophages (MH-S cells) were divided into six groups ($n = 3$ per group), including control, ovalbumin (OVA) + sh-NC, OVA + S100A8, OVA + S100A8, OVA + S100A9, OVA + S100A8 + LPS, and OVA + S100A9 + LPS groups. Lactic acid concentration (A) in cell supernatant was measured using a spectrophotometer. (B, C) In MH-S cells, concentration of phosphofructokinase (PFK) and hexokinase (HK) was measured using a spectrophotometer; they were increased in OVA group and can be inhibited by S100A8 or S100A9 knockdown. LPS intervention antagonist S100A8 or S100A9 knockdown's effect. (D-G) Glycolysis-related genes, GAPDH, HK2, LDHA, and PDH mRNA were detected by quantitative real-time PCR. They were a significant increase in MH-S cells with OVA induction, except for PDH, which showed decreased expression. S100A8 or S100A9 knockdown can inhibit GAPDH, HK2, and LDHA mRNA in OVA-induced MH-S cell injury models and increase PDH mRNA level, but LPS intervention can antagonist them. (H–L) Expression levels of TLR4, MyD88, TRIF, NF-κB, and p-ACLY/ACLY in MH-S cells were detected by Western blot. In MH-S cells with OVA induction, they were increased and S100A8 or S100A9 knockdown decreased them, while LPS intervention partially antagonized the effects of S100A8 or S100A9 knockdown. (M) Protein bands of Western blot. (mean ± standard deviation) **$p < 0.01$ compared to control group; ##$p < 0.01$ compared to OVA group; ^$p < 0.05$, ^^$p < 0.01$ compared to OVA + sh-S100A8 group; +$p < 0.05$, ++$p < 0.01$ compared to OVA + sh-S100A8 group.

them ($p < 0.01$) (Figs. 3A–3I). Mice with OVA-sensitized and -challenged had increased serum OVA-specific IgE levels ($p < 0.01$) (Fig. 3J), whereas S100A8 or S100A9 knockdown markedly decreased IgE levels compared to those in the OVA group ($p < 0.01$) (Fig. 3J). Histological analysis indicated an increase in peribronchial inflammatory infiltrates in the

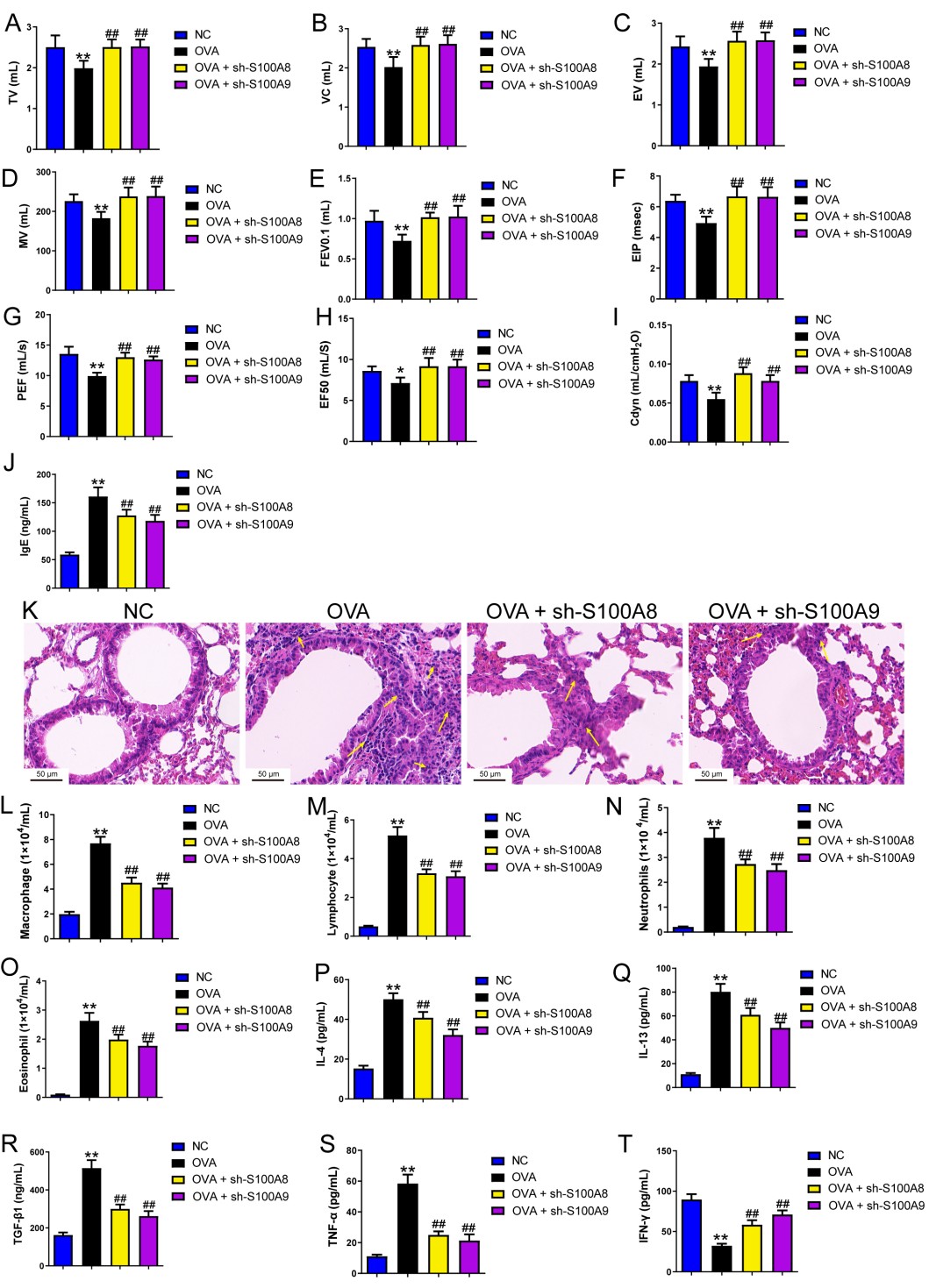

**Figure 3 S100A8/9 knockdown improved respiratory function, lung tissue injury and inflammation of broncho-alveolar lavage fluid in ovalbumin-sensitized and challenged mice.** BALB/C mice with sh-NC, sh-S100A8, or S100A9 intervention were divided into negative control (NC), ovalbumin (OVA), OVA + sh-S100A8, and OVA + sh-S100A9 groups ($n = 6$ per group). (continued on next page…)

**Figure 3 (…continued)**
(A–I) Observation of respiratory function was based on the detection of tidal volume (TV), vital capacity (VC), expiratory volume (EV), minute ventilation volume (MV), forced expiratory volume in 0.1 s (FEV0.1), end inspiratory pause (EIP), peak expiratory flow (PEF), mid expiratory flow (EF50), and dynamic lung compliance (Cdyn) in OVA mice; they were decreased in OVA rats compared to NC rats and S100A8 or S100A9 knockdown in mice with OVA administration increased them. (J) Serum OVA-specific IgE was detected by ELISA; in OVA + sh-S100A8 and OVA + sh-S100A9 groups, IgE was decreased compared to OVA group. (K) Hematoxylin-eosin staining observed S100A8 or S100A9 knockdown improved lung tissue damage (×400, Scare bar = 50 μm). The yellow arrow indicates representative inflammatory cell infiltration. In broncho-alveolar lavage fluid (BALF), (L) macrophages, (M) lymphocytes, (N) neutrophils, and (O) eosinophils were counted and were increased in OVA group. The concentration of IL-4 (P), IL-13 (Q), TGF-β1 (R), TNF-α (S), and IFN-g (T) in BALF were measured by ELISA. (mean ± standard deviation) $^{**}p < 0.01$ compared to NC group; $^{\#\#}p < 0.01$ compared to OVA group.

lungs of mice that were sensitized and challenged with OVA, whereas S100A8 or S100A9 knockdown markedly reduced it (Fig. 3K). Furthermore, macrophages, lymphocytes, neutrophils, and eosinophils were counted by Diff-Quik staining, and concentrations of IL-4, IL-13, TGF- β1, TNF-α, and IFN-γ were measured by ELISA. Most of them were significantly increased in the BALF from mice sensitized and challenged with OVA ($p < 0.01$) (Figs. 3L–3S) but the IFN-γ concentration was decreased ($p < 0.01$) (Fig. 3T). S100A8 or S100A9 knockdown antagonized the OVA-induced inflammatory response ($p < 0.01$) (Figs. 3L–3T). In summary, S100A8/9 knockdown improved respiratory function, lung tissue injury, and inflammation in mice sensitized and challenged with OVA.

## S100A8/9 knockdown suppressed macrophage polarization in OVA-sensitized and challenged mice

Changes in glycolysis *in vivo*. Flow cytometry was used to detect macrophage polarization in the BALF. The results showed that the proportions of M1 (CD86+) and M2 (CD206+) cells were increased in mice sensitized and challenged with OVA, whereas S100A8 or S100A9 knockdown decreased them ($p < 0.05$) (Figs. 4A, 4B). IHC was used to detect the expression of macrophage biomarkers. The total macrophage biomarkers CD68, M1 macrophage biomarker IRF-5, and M2 macrophage biomarker YM-1 were measured, and their positive cells increased in OVA-sensitized and challenged mice ($p < 0.01$) (Figs. 4C, 4D). However, S100A9 knockdown significantly decreased these levels in OVA-sensitized and challenged mice ($p < 0.05$) (Figs. 4C, 4D). Furthermore, the mRNA levels of IL-6, iNOS, Arg1, and IL-10 were determined using qRT-PCR. Among these, IL-6 and iNOS are genes related to M1 macrophages, whereas Arg1 and IL-10 are related to M1 macrophages (*Zhu et al., 2015*). They increased in OVA-sensitized sensitized and challenged mice, whereas S100A8 or S100A9 knockdown decreased them, except for IL-10, which showed the opposite trends ($p < 0.01$) (Figs. 4E–4F).

## S100A8/9 knockdown inhibited glycolysis in the lung of OVA-sensitized and challenged mice

Serum lactic acid concentration was markedly increased in the OVA group. However, S100A8 and S100A9 knockdown inhibited it ($p < 0.01$) (Fig. 5A), suggesting that glycolysis

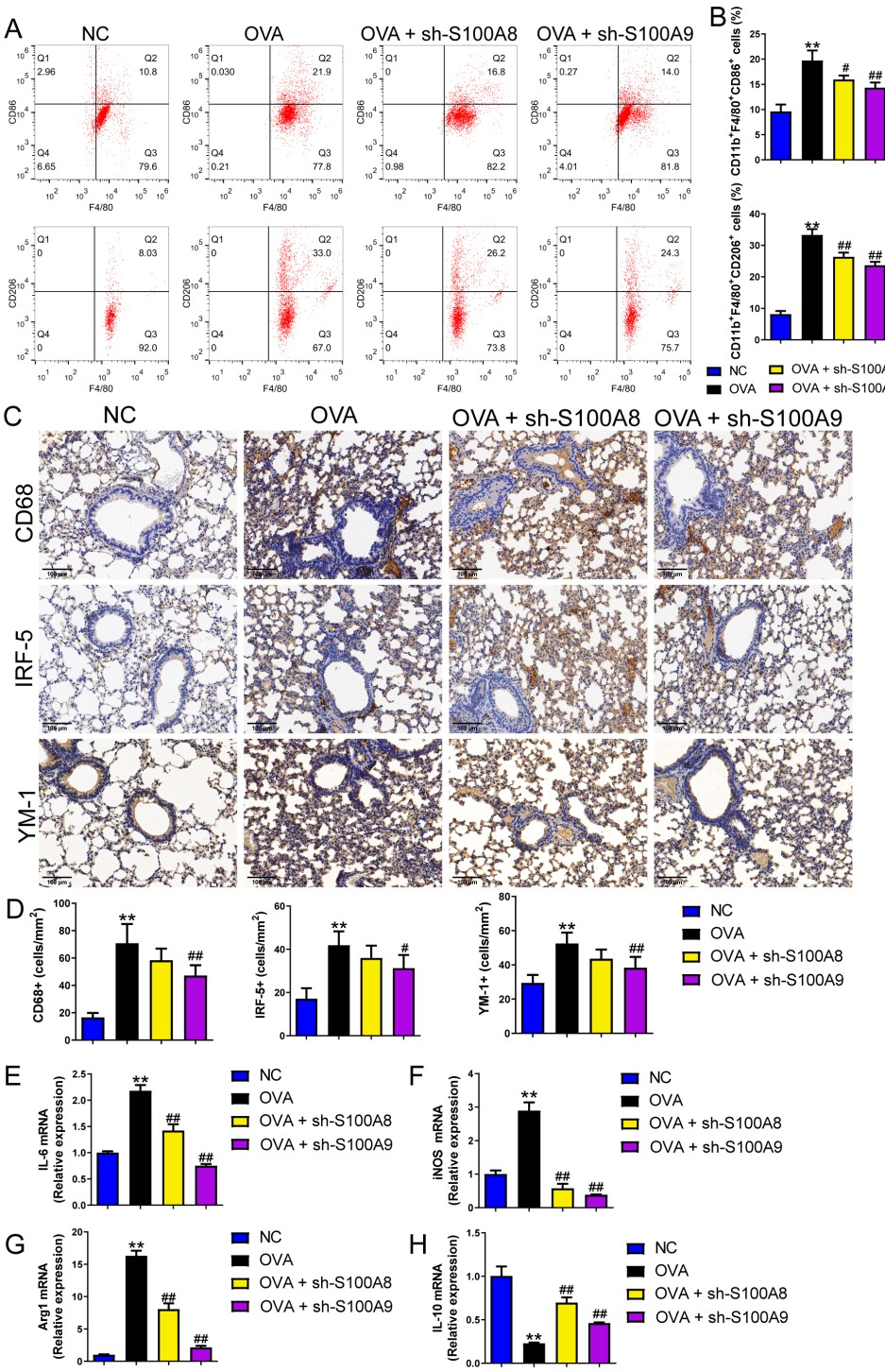

**Figure 4  S100A8/9 knockdown suppressed macrophage polarization in ovalbumin-sensitized and challenged mice.** BALB/C mice with sh-NC, sh-S100A8, or S100A9 intervention were divided into negative control (NC), ovalbumin (OVA), OVA + sh-S100A8, and OVA + sh-S100A9 groups ($n = 6$ per group). (A, B) Flow cytometry was used to detect polarization of 

**Figure 4 (…continued)**
macrophages in broncho-alveolar lavage fluid (BALF); the proportion of M1 (CD86+) and M2 (CD206+) cells was increased in mice with OVA administration, while in OVA-sensitized and challenged mice with S100A8 or S100A9 knockdown, they were decreased. (C) Immunohistochemistry was used to observe the expression levels of macrophage biomarkers. CD68 is a macrophage biomarker, IRF-5 is an M1 macrophage biomarker, and YM-1 is an M2 macrophage biomarker. (D) Statistic analysis of immunohistochemical image; CD68, IRF-5, and YM-1 positive cells were all increased in OVA-sensitized and challenged mice, but they were significantly decreased in OVA-sensitized sensitized and challenged mice with S100A9 knockdown. (E–H) The mRNA of IL-6, iNOS, Arg1 and IL-10 were detected by quantitative real-time PCR; they were increased in OVA-sensitized sensitized and challenged mice while S100A8 or S100A9 knockdown decreased them except for IL-10 which has opposite trends. (mean ± standard deviation) $^{**}p < 0.01$ compared to NC group; $^{#}p < 0.05$, $^{##}p < 0.01$ compared to OVA group.

was inhibited. Pyruvate dehydrogenase (PDH), lactate dehydrogenase (LDH) A, and HK2 are key enzymes involved in glycolysis (*Pereverzeva et al., 2022*). Their mRNAs increased in OVA-sensitized and challenged mice, whereas S100A8 and S100A9 knockdown significantly decreased their levels ($p < 0.01$) (Figs. 5A–5D). Furthermore, we used Western blot to measure the levels of LDHA, HK2, TLR4, MyD88, p-NF-κB/NF-κB, p-IκBα/IκBα, gasdermin D-N, and cleaved-caspase-1/caspase-1. OVA-induced allergic asthma increased all levels, whereas S100A8 and S100A9 knockdown decreased the levels in OVA-sensitized and challenged mice ($p < 0.01$) (Figs. 5E–5M).

## S100A9 overexpression had an adverse impact on respiratory function and lung tissue while enhancing inflammation in ovalbumin-sensitized and challenged mice

Evidence proved that the expression level of S100A8 is regulated by S100A9 (*Hobbs et al., 2003*). We found knockdown of S100A9 ameliorated injury in allergic mice and inhibited glycolysis in macrophages and lung tissues. To clarify the ability of S100A9 to regulate glycolysis, we used allergic asthmatic mice overexpressing S100A9 to observe the promotion of glycolysis by S100A9 overexpression. The glycolysis inhibitor, 3-BP, can inhibit HK2 (*Zhong et al., 2022*). Using 3-BP treatment, we explored the potential mechanisms by which S100A9 regulates glycolysis. S100A9 mRNA levels in the OVA + OE-S100A9 group were 2.94 times higher than those in the OVA group, and 3-BP intervention did not affect the mRNA levels ($p < 0.01$) (Fig. 6A). Serum OVA-specific IgE in OVA-sensitized and challenged mice with the 3-BP intervention was significantly decreased compared to that in the OVA group ($p < 0.01$), while S100A9 overexpression markedly increased IgE compared to that in the OVA group ($p < 0.01$) (Fig. 6B). Compared with the OVA group, TV, VC, EV, MV, FEV0.1, EIP, PEF, EF50, and Cdyn in the OVA + 3-BP group were markedly increased, whereas S100A9 overexpression increased them ($p < 0.05$) (Figs. 6C–6K). Histological analysis indicated a decrease in peribronchial inflammatory infiltrates in the lungs of OVA-sensitized and challenged mice with 3-BP intervention, while S100A9 overexpression enhanced lung tissue damage (Fig. 6L). Furthermore, in the BALF of the OVA + 3-BP group compared to that of the OVA group, macrophages, lymphocytes, neutrophils, eosinophils, and the concentrations of IL-4, IL-13, TGF-β1, and TNF-α were significantly decreased ($p < 0.01$), while their levels were increased in

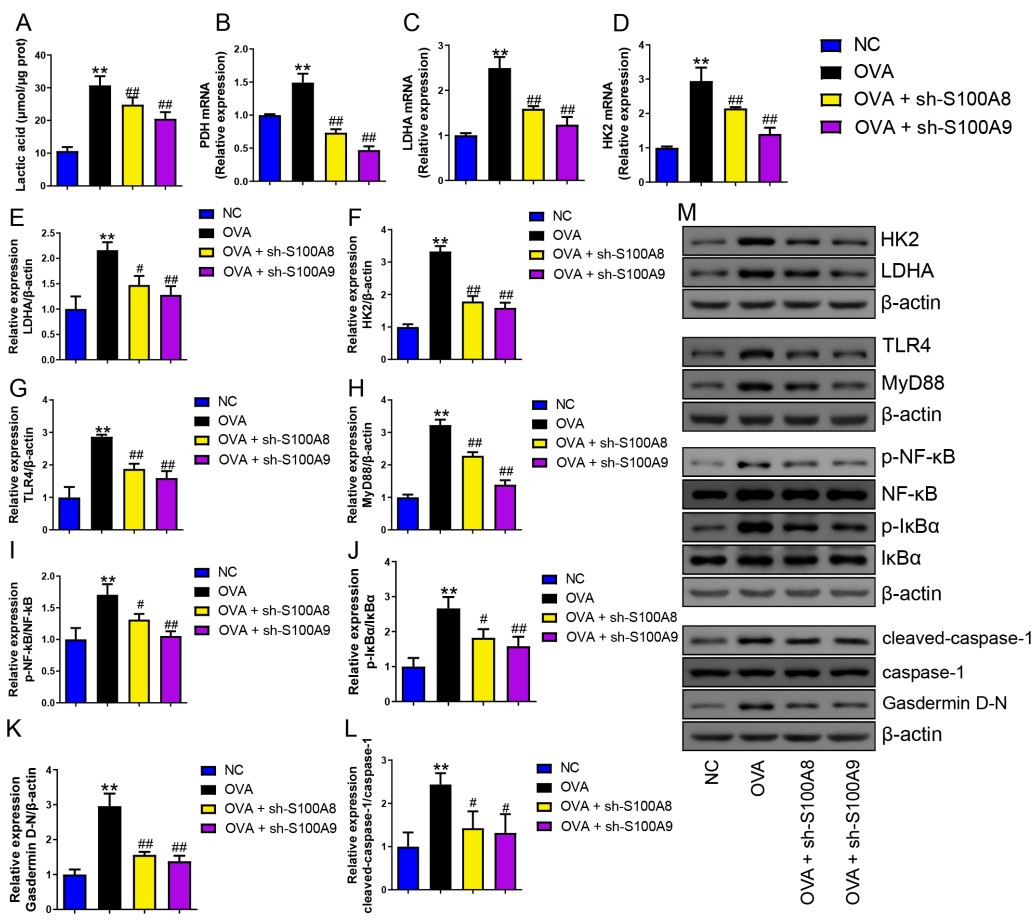

**Figure 5** **S100A8/9 knockdown inhibited glycolysis in ovalbumin-sensitized and challenged mouse lung tissue.** BALB/C mice with sh-NC, sh-S100A8, or S100A9 intervention were divided into negative control (NC), ovalbumin (OVA), OVA + sh-S100A8, and OVA + sh-S100A9 groups. (A) Serum lactic acid concentration was measured by using a spectrophotometer ($n = 6$). (B–D) Pyruvate dehydrogenase (PDH), lactate dehydrogenase (LDH) A, and hexokinase (HK) 2 are key enzymes of glycolysis, and mRNA of them were increased in mice post-OVA-challenged, while they were decreased in OVA-sensitized and challenged mice by S100A8 and S100A9 knockdown ($n = 3$). Quantitative real-time PCR is used for mRNA measurement. (E–M) Western blot is used for measurement of LDHA, HK2, TLR4, MyD88, p-NF-kB/NF-κB, p-IκBα/IκBα, Gasdermin D-N, and cleaved-caspase-1/caspase-1 levels ($n = 3$). They were all increased by OVA-sensitized and challenged, and were decreased in OVA-sensitized and challenged mice by S100A8 and S100A9 knockdown. (mean ± standard deviation) $^{**}p < 0.01$ compared to NC group; $^{\#}p < 0.05$, $^{\#\#}p < 0.01$ compared to OVA group.

OVA-sensitized and challenged mice with S100A9 overexpression ($p < 0.01$) (Figs. 6M–6T). However, compared to the OVA group, IFN-γ concentration was increased in the BALF of the OVA + 3-BP group, whereas it was decreased in the OE-S100A9 group ($p < 0.01$) (Fig. 6U). Furthermore, 3-BP treatment antagonized the effects of S100A9 overexpression on inflammation. S100A9 overexpression had detrimental effects on respiratory function and lungs in OVA-sensitized and -challenged mice, while also enhancing inflammation by promoting glycolysis.

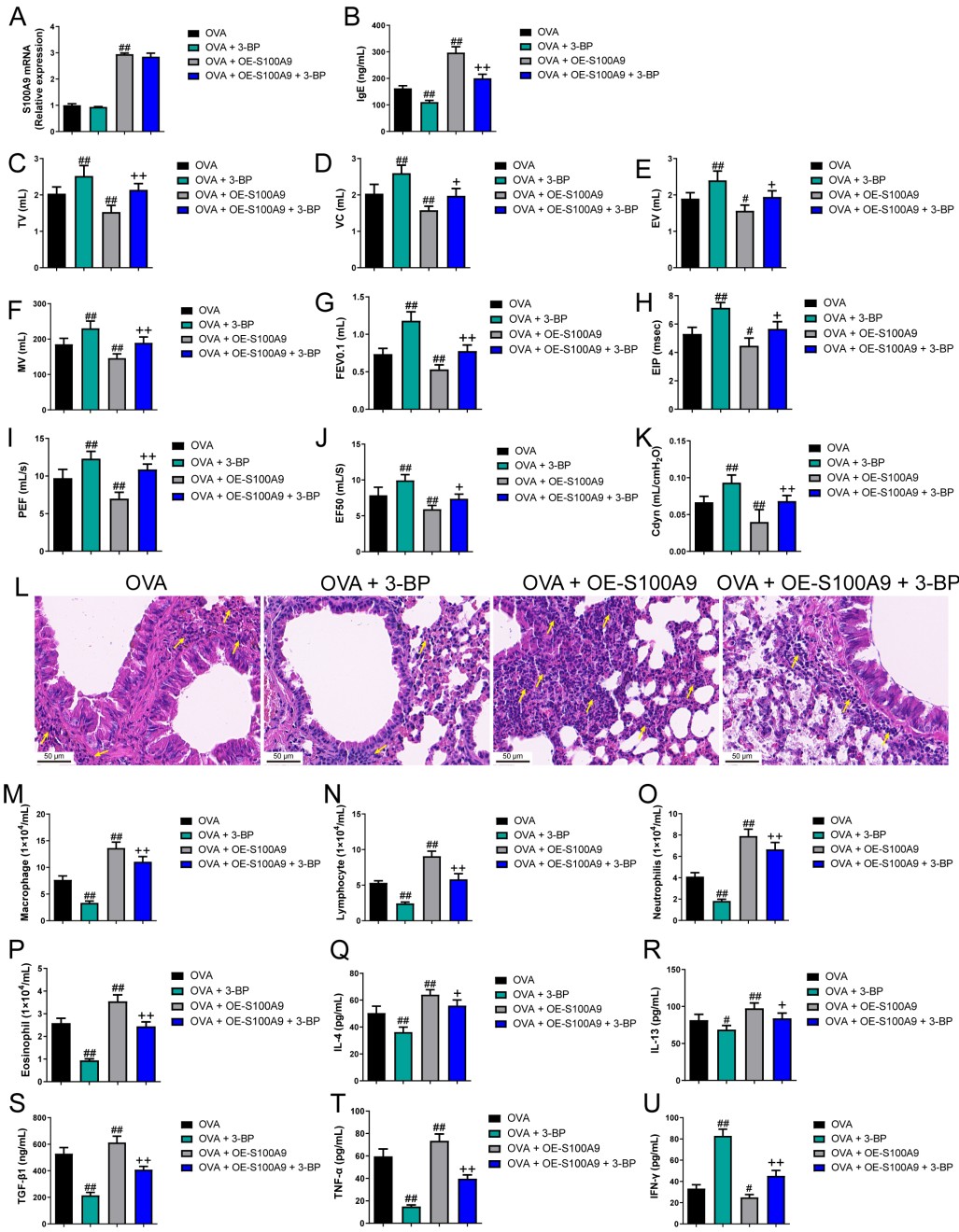

**Figure 6** **S100A9 overexpression had an adverse impact on respiratory function and lung tissue while enhancing inflammation in ovalbumin-sensitized and challenged mice.** Ovalbumin (OVA) sensitized BALB/C mice were divided into groups receiving 3-bromopyruvate (3-BP), S100A9 plasmid intervention, or both. The resulting groups were OVA, OVA + 3-BP, OVA + OE-S100A9, and OVA + OE-S100A9 + 3-BP ($n = 6$ per group). (A) S100A9 mRNAs were detected by quantitative real-time PCR and were increased in OVA + OE-S100A9 and OVA + OE-S100A9 + 3-BP group. (B) Serum OVA-specific IgE was detected by ELISA; (continued on next page...)

**Figure 6 (...continued)**
compared to OVA group, IgE was decreased in OVA + 3-BP groups and increased in OE-S100A9 group. (C–K) Observation of respiratory function was based on the detection of tidal volume (TV), vital capacity (VC), expiratory volume (EV), minute ventilation volume (MV), forced expiratory volume in 0.1 s (FEV0.1), end inspiratory pause (EIP), peak expiratory flow (PEF), mid expiratory flow (EF50), and dynamic lung compliance (Cdyn) in OVA mice; they were increased in OVA + 3-BP group compared to OVA group and S100A9 overexpression in mice with OVA administration decreased them. (L) Hematoxylin-eosin staining observed S100A9 overexpression enhanced lung tissue damage (×400, Scare bar = 50 μm) while 3-BP can antagonize it. The yellow arrow indicates representative inflammatory cell infiltration. In broncho-alveolar lavage fluid from OVA-sensitized and challenged mice, (M) macrophages, (N) lymphocytes, (O) neutrophils, and (P) eosinophils were counted. (Q–U) Concentration of IL-4, IL-13, TGF-β1, TNF-α and IFN-γ in broncho-alveolar lavage fluid was measured by ELISA. Inflammatory cells and cytokines were decreased in OVA + 3-BP group, but they were increased in OVA + OE-S100A9 group. In addition, 3-BP can antagonize the effect of S100A9 overexpression. (mean ± standard deviation) $^\#p < 0.05$, $^{\#\#}p < 0.01$ compared to OVA group; $^+p < 0.05$, $^{++}p < 0.01$ compared to OVA + OE-S100A9 group.

### S100A9 overexpression promoted macrophage polarization in ovalbumin-sensitized and challenged mice

Flow cytometry results showed that the proportion of M1 (CD86+) and M2 (CD206+) macrophages was decreased in OVA-induced allergic asthma mice with 3-BP intervention, whereas they increased in OVA-sensitized and challenged mice with S100A9 overexpression ($p < 0.01$) (Figs. 7A, 7B). Immunohistochemistry was performed to observe the expression levels of macrophage biomarkers. CD68 is a macrophage biomarker, IRF-5 is an M1 macrophage biomarker, and YM-1 is an M2 macrophage biomarker. CD68, IRF-5, and YM-1 positive cells were all decreased in OVA-sensitized and challenged mice with 3-BP administration, but were significantly increased after S100A9 overexpression ($p < 0.01$) (Fig. 7D). In the lungs, IL-6, iNOS, and Arg1 mRNA levels decreased in OVA-sensitized and challenged mice with 3-BP intervention and increased in OVA-sensitized and challenged mice with S100A9 overexpression, whereas IL-10 showed the opposite trend. Furthermore, results showed that 3-BP intervention antagonized the stimulatory effects of S100A9 overexpression on macrophage polarization in OVA-sensitized and -challenged mice ($p < 0.01$) (Fig. 7).

### S100A9 overexpression inhibited glycolysis in the lung of ovalbumin-sensitized and challenged mice

Compared to the OVA group, the serum lactic acid concentration in OVA-sensitized and -challenged mice was inhibited by 3-BP intervention and were increased in the OVA + OE-S100A9 group ($p < 0.01$) (Fig. 8A). PDH, LDH, and HK2, the key enzymes of glycolysis, were measured by qRT-PCR and were decreased in mice post-OVA challenge with 3-BP intervention, whereas they were increased in OVA-sensitized and challenged mice by S100A9 overexpression ($p < 0.05$) (Figs. 8B–8D). The expression levels of LDHA, HK2, TLR4, MyD88, p-NF-κB/NF-κB, p-IκBα/IκBα, gasdermin D-N, and cleaved-caspase-1/caspase-1 levels were measured by Western blot. Compared to the OVA-sensitized and challenged mice, they all decreased in the OVA + 3-BP group and increased in the OVA + OE-S100A9 group ($p < 0.01$) (Figs. 8E–8L). Additionally, 3-BP antagonized the promotion of S100A9 overexpression in serum lactic acid, glycolysis-related enzymes, and genes.

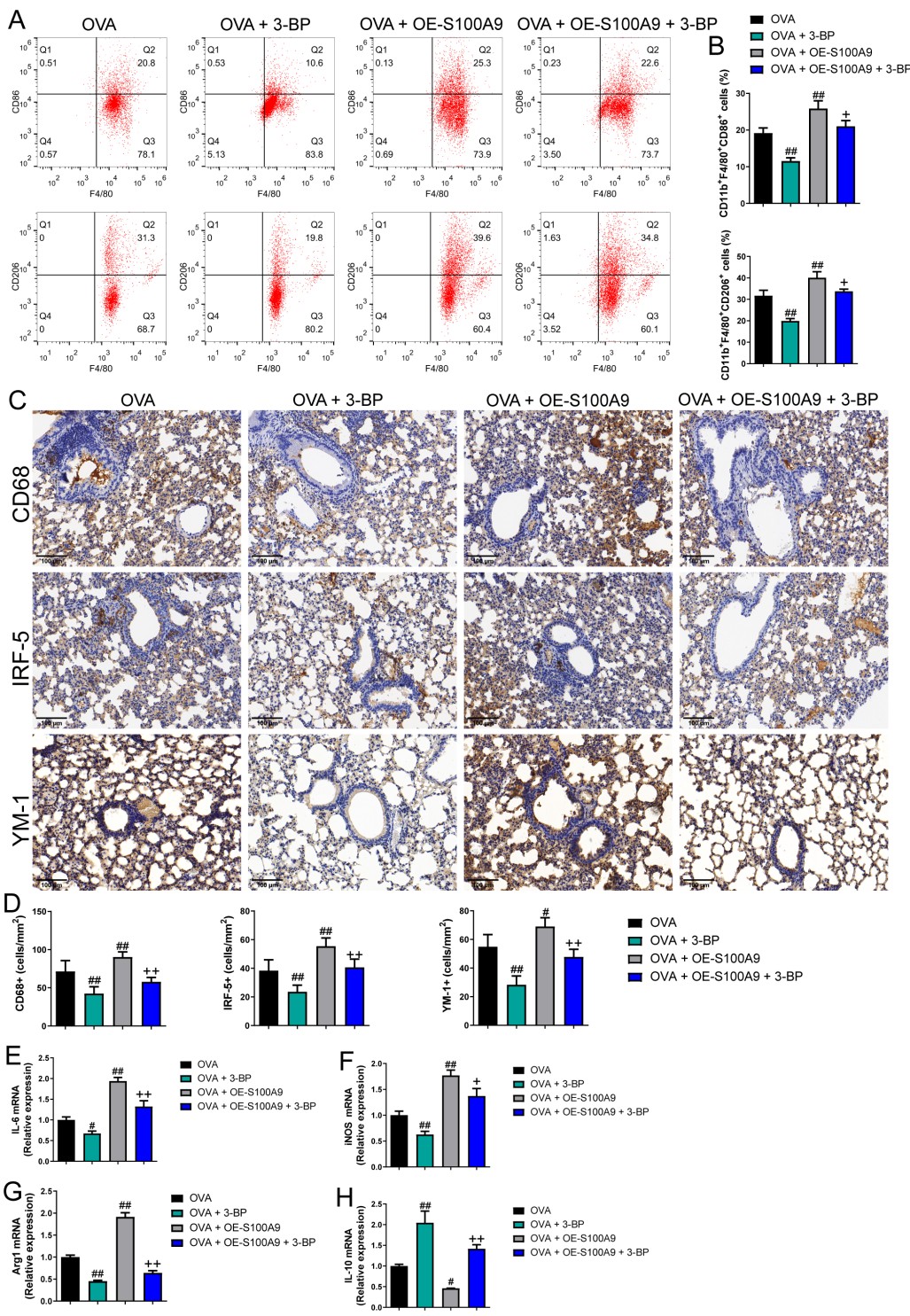

**Figure 7** **S100A9 overexpression promoted macrophage polarization in ovalbumin-sensitized and challenged mice.** BALB/C mice are sensitized and challenged with ovalbumin (OVA). They were divided into groups receiving 3-bromopyruvate (3-BP), S100A9 plasmid intervention, or both. The resulting groups were OVA, OVA + 3-BP, OVA + OE-S100A9, (continued on next page...)

**Figure 7 (…continued)**
and OVA + OE-S100A9 + 3-BP ($n = 6$ per group). (A, B) Flow cytometry was used to detect polarization of macrophages in broncho-alveolar lavage fluid (BALF); compared to OVA-sensitized and challenged mice, the proportion of M1 (CD86+) and M2 (CD206+) macrophages was decreased in mice with OVA and 3-BP administration, while in OVA-sensitized and challenged mice with S100A9 overexpression, they were increased. (C) Immunohistochemistry was used to observe the expression levels of macrophage biomarkers. CD68 is a macrophage biomarker, IRF-5 is an M1 macrophage biomarker, and YM-1 is an M2 macrophage biomarker. (D) Statistic analysis of immunohistochemical image; CD68, IRF-5, and YM-1 positive cells were all decreased in OVA-sensitized and challenged mice with 3-BP administration, but they were significantly increased after S100A9 overexpression. (E–H) In the lung, the mRNA of IL-6, iNOS, Arg1, and IL-10 were detected by quantitative real-time PCR; IL-6, iNOS and Arg1 were decreased in OVA-sensitized and challenged mice with 3-BP intervention and increased in OVA-sensitized and challenged mice with S100A9 overexpression, while IL-10 has the opposite trend. (mean ± standard deviation) #$p < 0.05$, ##$p < 0.01$ compared to OVA group; +$p < 0.05$, ++$p < 0.01$ compared to OVA + OE-S100A9 group.

## DISCUSSION

Allergic asthma is a chronic lung disease characterized by reversible airway obstruction resulting in airflow limitation, as well as physiological symptoms such as wheezing, coughing, and changes in the bronchial structure (*Hough et al., 2020*). Macrophage polarization plays an important role in the development of allergic asthma. The *in vivo* results of our study demonstrated that the knockdown of S100A8 or S100A9 inhibited M1 and M2 macrophage polarization and improved respiratory function and lung injury in mice with allergic asthma. In particular, S100A9 overexpression exacerbates lung injury and inflammation in allergic asthma. Additionally, the glycolysis inhibitor 3-BP (HK2 inhibitor) antagonized S100A9 overexpression, effects, suggesting that the regulation of glycolysis plays a critical role in the involvement of S100A9 in allergic asthma and HK2 might be a target of S100A9. The OVA-induced asthma mouse model is a classic allergic asthma model that usually shows decreased IFN-γ levels (*Lertnimitphun et al., 2021*). IFN-γ is mainly secreted by T lymphocytes, macrophages, mucosal epithelial cells, or natural killer cells (*Piao et al., 2023*; *Ding et al., 2022*). IFN-γ can also activate M1 macrophages (*Fu et al., 2023*), which contradicts the results we observed *in vivo*, suggesting IFN-γ levels in BALF are not dependent on macrophages. Knockdown of S100A8 or S100A9 increased IFN-γ levels, suggesting that S100A proteins may be involved in immune imbalance. This could be an interesting direction for future research.

We found that S100A8 and S100A9 knockdown inhibited M1 and M2 macrophage polarization. M1 macrophages express pro-inflammatory factors, such as IL-6, IL-1β, and TNF-α, as well as iNOS, and play a role in recruiting and activating other immune cells (*Saradna et al., 2018*). In contrast, M2 macrophages express Ym1, Arg1, CD206, IL-10, TGF-β, and other anti-inflammatory factors (*Müller et al. 2007*; *Dewhurst et al., 2017*). There are four subtypes of M2 macrophages which express different cytokines, chemokines, and growth factors, and have varying functions in inflammation (*Murray et al., 2014*; *Roszer, 2015*). Among them, M2a macrophages are associated with the lung allergic inflammation that induced by Th2 polarizing cytokines (IL-4 or IL-13) to overexpression of IL-10, TGF-β, and inflammatory chemokines; moreover, M2b macrophages are facilitated by immune complexes and have been demonstrated to play an important role in the

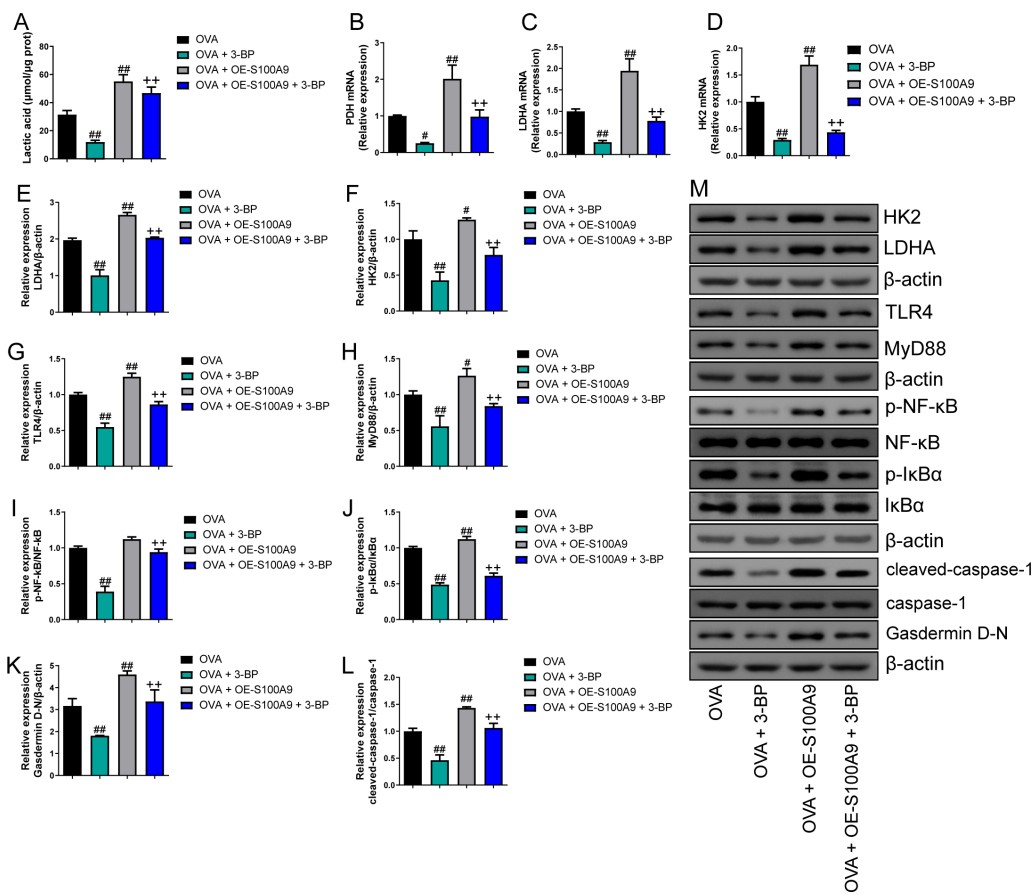

**Figure 8** **S100A9 overexpression inhibited glycolysis in ovalbumin-sensitized mouse lung tissue.**
Ovalbumin (OVA) sensitized BALB/C mice were divided into groups receiving 3-bromopyruvate
(3-BP), S100A9 plasmid intervention, or both. The resulting groups were OVA, OVA + 3-BP, OVA +
OE-S100A9, and OVA + OE-S100A9 + 3-BP. (A) Serum lactic acid concentration was measured by using
a spectrophotometer ($n = 6$). (B–D) Pyruvate dehydrogenase (PDH), lactate dehydrogenase (LDH) A and
hexokinase (HK) 2 are key enzymes of glycolysis and mRNAs of them were decreased in mice post-OVA-
challenged with 3-BP intervention, while they were increased in OVA-sensitized and challenged mice
by S100A9 overexpression ($n = 3$). Quantitative real-time PCR is used for mRNA measurement. (E-M)
Western blot is used for measurement of LDHA, HK2, TLR4, MyD88, p-NF-kB/NF-κB, p-IκBα/IκBα,
Gasdermin D-N, and cleaved-caspase-1/caspase-1 levels ($n = 3$). Compared to OVA-sensitized and
challenged mice, they were all decreased in OVA-sensitized mice by 3-BP intervention and increased in
OVA-sensitized and challenged mice with S100A9 overexpression. In addition, 3-BP can antagonize the
promotion of S100A9 overexpression on glycolysis. (mean ± standard deviation) $^{\#}p < 0.05$, $^{\#\#}p < 0.01$
compared to OVA group; $^{++}p < 0.01$ compared to OVA + OE-S100A9 group.

Th2 immune response (*Ross, Devitt & Johnson, 2021*). Additionally, the subtype M2b
macrophages exhibit a phenotype similar to that of M1 macrophages (IL-1β activates
and releases TNF-α), simultaneously secreting high levels of IL-10 (*Kang et al., 2022*).
However, exosomes derived from M2b macrophages antagonize colitis (*Yang et al., 2019*).
The role of the M2b macrophages in allergic asthma is not yet clear. However, IL-10 is
an anti-inflammatory cytokine that ensures mitochondrial integrity and inhibits cellular
glycolysis levels (*Ip et al., 2017*). Low levels of IL-10 or few M2b macrophages may be

involved in regulating macrophage metabolism and promoting the development of allergic asthma. Further experiments are needed to explore the macrophage phenotype associated with IL-10 expression. Macrophage polarization is a complex process involving cellular activities. The pro-inflammation effect of M1 macrophages and the anti-inflammation effect of M2 macrophages are well known. Recent evidence suggests that M1 and M2 macrophages can coexist, with changes in cellular metabolism to regulate their polarization (*Tang et al., 2023*). An OVA-induced macrophage injury model showed the coexistence of M1 and M2 macrophages (*Wo et al., 2023*), indicating that macrophage polarization may promote OVA-induced lung damage. We also found that knockdown of S100A8 or S100A9 inhibited lactate acid and LDHA levels. Studies have shown that increased glycolysis generally manifests as increased lactate acid and LDHA (*Zhang et al., 2023*). Our study highlighted the potential of inhibiting S100A8 and S100A9 to ameliorate allergic asthma by stabilizing macrophage polarization and inhibiting glycolysis.

In the last three years, immunometabolic studies on allergies have reported that asthma is associated with increased aerobic glycolysis (*Goretzki et al., 2023*). Studies have found that serum lactate acid levels in patients with asthma are significantly higher than those in healthy controls, indicating the presence of glycolytic reprogramming of glycolysis in asthma (*Ostroukhova et al., 2012*). The competitive glucose inhibitor 2-DG inhibits Arg2 expression *in vitro* (*Svedberg et al., 2019*). The knockdown of S100A8 or S100A9 inhibited glycolysis and M2 polarization, which is consistent with previous findings and further highlights the importance of glycolysis in macrophage polarization.

HK2 is a key enzyme involved in glucose phosphorylation (the first step in the glucose metabolism pathway), and evidence has shown that inhibiting HK2 transcription reduces glycolysis in macrophages (*Yang et al., 2022*; *Yuan et al., 2022*). S100A8 and S100A9 knockdown inhibited HK2 and GAPDH expression, suggesting that S100A8 and S100A9 are involved in the regulation of glycolysis in allergic asthmatic macrophages. S100A8 is an endogenous ligand for TLR4, which has been shown to induce intracellular translocation of MyD88 as well as NF-κB activation to promote the elevation of TNF-α levels (*Vogl et al., 2007*). Additionally, LPS-induced TLR4 activation regulates metabolic fluxes, which is mainly dominated by the enhancement of histone acetylation resulting from the production of acetyl-CoA from glucose (*Lauterbach et al., 2019*). Early TLR4-driven aerobic glycolysis was initiated by overlapping and redundant contributions of MyD88- and TRIF-dependent signaling pathways, as well as downstream mTOR activation (*Fensterheim et al., 2018*). An increase in glucose uptake and accelerated glycolytic flux promote mitochondrial citrate production and mitochondrial citrate can be converted to acetyl-CoA by ACLY (*Granchi, 2018*). The TLR4/MyD88/TRIF/NF-κB signaling pathway and ACLY activity were inhibited in OVA-induced MH-S cells with knockdown of S100A8 or S100A9 and allergic asthma mice, suggesting that knockdown of S100A8 or S100A9 inhibits glycolysis by suppressing the TLR4/MyD88/TRIF/NF-κB signaling pathway. LPS intervention antagonizes the effects of S100A8 or S100A9 knockdown in OVA-induced MH-S cells. In addition, 3-BP, an inhibitor of HK-II, antagonized the effects of S100A9 overexpression, suggesting that HK-II is a key gene involved in the S100A8 and S100A9 regulation of glycolysis in mice with allergic asthma, which warrants further investigation. In addition, we observed the effect

of S100A8 or S100A9 knockdown on pyroptosis. The inhibition of macrophage pyroptosis is usually beneficial in reducing inflammation (*Sun et al., 2021*). However, studies have shown that inhibition of glycolysis inhibits pyroptosis, and there is still a lack of research on the role and mechanism of macrophage pyroptosis in allergic asthma (*Zasłona et al., 2020*; *Aki et al., 2022*).

Our study has certain limitations owing to the complexity of macrophage polarization and function. However, research on macrophage subtypes remains insufficient. Further investigation is necessary to differentiate the effects of S100A8/A9 on the proportion of macrophage subtypes in allergic asthma. Our study used only OVA to construct models *in vitro* and *in vivo*. Although OVA was used to create a classical allergic asthma model, it could not generalize all clinical manifestations; therefore, future validation experiments on multiple allergic asthma models are needed to strengthen our hypothesis. Furthermore, as basic research, this result cannot be generalized to clinical practice and still requires a substantial number of animal experiments and clinical data. Overall, our work provides new ideas and directions for the treatment of allergic asthma.

## CONCLUSIONS

Our study highlights that S100A8 and S100A9 play critical roles in the pathogenesis of allergic asthma by promoting macrophage perturbation and glycolysis through the TLR4/MyD88/NF-κB signaling pathway. Inhibition of S100A8 and S100A9 may be a potential therapeutic strategy for allergic asthma. Further basic and human studies are required to explore the underlying mechanisms.

### Funding

The Medical Health Science and Technology Project of Zhejiang Provincial Health Commission (2024KY449) supported the APC for this article. The funders had no role in study design, data collection and analysis, decision to publish, or preparation of the manuscript.

### Grant Disclosures

The following grant information was disclosed by the authors:
Medical Health Science and Technology Project of Zhejiang Provincial Health Commission: 2024KY449.

### Competing Interests

The authors declare there are no competing interests.

### Author Contributions

- Xiaoyi Ji conceived and designed the experiments, authored or reviewed drafts of the article, responsible for submission, and approved the final draft.

- Chunhua Nie analyzed the data, prepared figures and/or tables, and approved the final draft.
- Yuan Yao performed the experiments, prepared figures and/or tables, and approved the final draft.
- Yu Ma performed the experiments, prepared figures and/or tables, and approved the final draft.
- Huafei Huang analyzed the data, prepared figures and/or tables, and approved the final draft.
- Chuangli Hao conceived and designed the experiments, authored or reviewed drafts of the article, responsible for submission, and approved the final draft.

### Animal Ethics

The following information was supplied relating to ethical approvals (i.e., approving body and any reference numbers):

The Animal Experimentation Ethics Committee of Zhejiang Eyong Pharmaceutical Research and Development Center [SYXK(Zhe)2021-0033; approval number: ZJEY-20221205-02)

### Data Availability

The raw measurements can be found in Files S1 and S2.

### Supplemental Information

Supplemental information for this article can be found online at http://dx.doi.org/10.7717/peerj.17106#supplemental-information.

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
