# Peer review of "S100A8/9 modulates perturbation and glycolysis of macrophages in allergic asthma mice"

_PeerJ, doi:10.7717/peerj.17106_

## Round 0.1 · original submission · Minor Revisions

Dear Dr. Hao, the reviewers have raised just some minor comments and are globally agree to publish your manuscript after the revision of the text following their comments.

**Language Note:** The review process has identified that the English language must be improved. PeerJ can provide language editing services - please contact us at copyediting@peerj.com for pricing (be sure to provide your manuscript number and title). Alternatively, you should make your own arrangements to improve the language quality and provide details in your response letter. – PeerJ Staff

Reviewer 1 ·

Basic reporting

The manuscript requires some English grammatical revisions throughout the entire text. Research idea presents an innovative approach to elucidate the mechanism of S100A8/9 in perturbation and glycolysis of macrophages in allergic asthma mice, the authors had good supporting data to address and answer the rationale behind this research.

1. Grammar- needs revisions, through the whole manuscript. Please see remarks in the annotated text.
Abstract lines 16-46 needs to be rewritten, to simplify text, to make it more coherent to the reader. Please see comments in the annotated text.
2. Introduction will benefit from a description of the markers used to differentiate polarized macrophages, explain what each marker stands for (full name+ abbreviation) and refer to literature why the marker is important for in asthma (line 69).
3. Hypothesis, state your hypothesis and what you hope to find, expand on the importance of this research (line 120).

Experimental design

Within the Aims and Scope of the journal, with good primary research questions, would benefit from clear Hypothesis statement.
4. Line 163, stat which kits are used, or refer to the section they are describes, or delete from this section.
5. Results will benefit from removing duplication or literature heavy sections that should be included in the introduction. make sure to state results in a concise and clear matter, w/o adding too much background information.
6. Line 242-247, 294-299, no need to explain knock down clones in details, better to deliver results that are directly contributing to the research questions. QC results should be noted in short i.e., “Knocked down clones were created”, remove the result from the main figure and add it to a supplementary figure section.
7. Figure 3M+ Figure 6L, add arrows and enlarged images to demonstrate immune infiltrate.
8. Line 314, low IFN-g decrease, try to hypothesis in the discussion what is the meaning of this result. How does it relate to your research question?
9. Discussion will benefit from suggest a mechanism that might explain the phenomena observed (Line 411, 425).
10. Line 460, Have you tried to knock down these pathways and see similar results? might make your case stronger.

Validity of the findings

The conclusions are well-articulated, directly tied to the original research question, and effectively supported by relevant literature. The manuscript displays a strong foundation but can benefit from the outlined improvements to enhance clarity.

Annotated reviews are not available for download in order to protect the identity of reviewers who chose to remain anonymous.

Reviewer 2 ·

Basic reporting

The paper "S100A8/9 modulates perturbation and glycolysis of macrophages in allergic asthma mice" focuses on the role of S100A8 and S100A9 in allergic asthma, particularly their effects on macrophage polarization and glycolysis. The language is clear and professional, suitable for a scientific paper. The paper seems well-structured with a clear focus on exploring the roles of specific proteins in asthma, potentially contributing new knowledge to the field.

Experimental design

The study uses both cell models and mouse models to investigate the effects of S100A8/A9. It employs various techniques like enzyme-linked immunosorbent assay, quantitative PCR, flow cytometry, and Western blot. This suggests a comprehensive and logical approach to validate the findings.
Here are some potential issues and areas for improvement:
1. Is there a dose-dependent effect of S100A8/A9 on macrophage behavior and glycolysis?
2. Please provide the gating strategy for flow cytometric analysis (Figure 5A, 8A) in the supplementary materials.
3. While the study provides some mechanistic insights, a deeper exploration of the underlying molecular pathways could enhance the understanding of how S100A8/A9 affects macrophage behavior and glycolysis.
4. The methods section could benefit from more detail, especially regarding the experimental conditions and parameters.

Validity of the findings

The study appears to add new insights into the role of S100A8/A9 in allergic asthma, particularly in the context of macrophage polarization and glycolysis. A more extensive review of related studies could be included to better position the research within the current state of knowledge.
Here are some potential issues and areas for improvement:
1. The paper could more explicitly address its limitations, such as potential biases, the limitations of the mouse model, and the extrapolation of results to human asthma.
2. The paper could further discuss the potential therapeutic implications of the findings, including how they might translate into clinical practice or lead to new treatments.

Additional comments

no comment

---

## Round 0.2 · accepted · Accept

Dear Dr. Hao, the reviewers have returned positive comments on your revised manuscript.

Reviewer 1 ·

Basic reporting

The authors' effective integration of the suggested corrections has significantly boosted the clarity and understanding of both the manuscripts and the accompanying data. These revisions not only improve the overall quality of the material but also deepen our comprehension of the causes underlying asthma, thereby offering valuable insights for researchers and clinicians alike.

Experimental design

The authors have successfully incorporated the recommended corrections

Validity of the findings

The authors have successfully incorporated the recommended corrections

Additional comments

The authors' effective integration of the suggested corrections has significantly boosted the clarity and understanding of both the manuscripts and the accompanying data. These revisions not only improve the overall quality of the material but also deepen our comprehension of the causes underlying asthma, thereby offering valuable insights for researchers and clinicians alike.